# Extreme synchronization transitions

Seungjae Lee ©[1] ✉, Lennart J. Kuklinski[1] & Marc Timme ©[1,2,3,4] ✉

Across natural and human-made systems, transition points mark sudden changes of order and are thus key to understanding overarching system features. Motivated by recent experimental observations, we here uncover an intriguing class of transitions in coupled oscillators, extreme synchronization transitions, from asynchronous disordered states to synchronous states with almost completely ordered phases. Whereas such a transition appears like discontinuous or explosive phase transitions, it exhibits markedly distinct features. First, the transition occurs already in finite systems of $N$ units and so constitutes an intriguing bifurcation of multi-dimensional systems rather than a genuine phase transition that emerges in the thermodynamic limit $N \to \infty$ only. Second, the synchronization order parameter jumps from moderate values of the order of $N^{-1/2}$ to values extremely close to 1, its theoretical maximum, immediately upon crossing a critical coupling strength. We analytically explain the mechanisms underlying such extreme transitions in coupled complexified Kuramoto oscillators. Extreme transitions may similarly occur across other systems of coupled oscillators as well as in certain percolation processes. In applications, their occurrence impacts our ability of ensuring or preventing strong forms of ordering, for instance in biological and engineered systems.

Transition points mark the qualitative change of collective phenomena upon varying system parameters, often between a less and a more ordered system state, with broad applications in physics, biology, engineering and beyond, see, e.g.,[1–9]. Distinguishing different classes of transitions and clarifying their underlying mechanisms are essential because the class of transition significantly impacts our ability to predict and understand state changes.

Phase transitions emerge in the structure and dynamics of complex systems in the thermodynamic limit of infinitely many units, $N \to \infty$. Whereas continuous phase transitions, such as to ferromagnetic order in spin systems, imply a smooth change in the degree of emergent order, discontinuous transitions such as the freezing of water, induce a jump in the degree of order[1].

Synchronization, the temporal ordering of phases of coupled oscillatory units, constitutes a paradigmatic ordering process emerging in nonlinear dynamical systems[10,11]. It stands as a temporal analog of structural ordering processes such as the emergence of ferromagnetism or freezing in many-particle systems [12]. The Kuramoto model mathematically captures key aspects of synchronization processes of coupled limit cycle oscillators, exhibiting a number of intriguing properties[13–16]. Specifically, systems of Kuramoto phase oscillators with natural frequencies drawn from a unimodal distribution exhibit a continuous phase transition to frequency locking and ultimately phase-locking with increasing the coupling strength $K \in \mathbb{R}$ (Fig. 1a). The synchronization order parameter

$$r = \left| \frac{1}{N} \sum_{\nu=1}^{N} e^{i x_\nu} \right| \tag{1}$$

quantifies the degree of coherence of the oscillators' phase state variable $x_\nu \in (-\pi, \pi]$ for $\nu \in [N] := \{1, 2, \cdots, N\}$. The phase transition to synchrony constitutes a transition of qualitative state change at a defined coupling strength $K_c$ emerging in the thermodynamic limit as the number of units $N \to \infty$. In contradistinction, finite-$N$ systems exhibit a crossover regime where ordering gradually increases in a range of coupling strengths $K$, see also Fig. 1a. Moreover, coupled

[1]Chair for Network Dynamics, Institute of Theoretical Physics and Center for Advancing Electronics Dresden (cfaed), Technische Universität Dresden, 01062 Dresden, Germany. [2]Cluster of Excellence Physics of Life, Technische Universität Dresden, 01062 Dresden, Germany. [3]Center Synergy of Systems, Technische Universität Dresden, 01062 Dresden, Germany. [4]Lakeside Labs, 9020 Klagenfurt, Austria. ✉e-mail: seungjae.lee@tu-dresden.de; marc.timme@tu-dresden.de

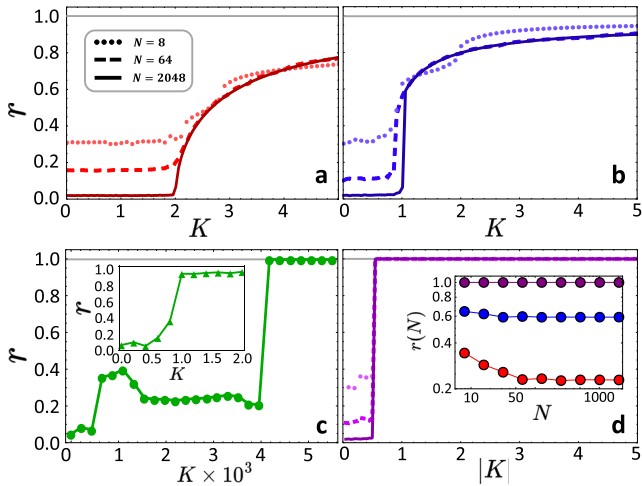

**Fig. 1 | From discontinuous to extreme synchronization transitions.** Panels show classical Kuramoto order parameter (1) as a function of coupling strength. **a** Continuous synchronization phase transition in the Kuramoto model with unimodal natural frequency distribution. **b** Discontinuous synchronization phase transition in the Kuramoto model with bimodal natural frequency distribution. Phase transitions with defined transition point $K_c$ in (**a**) and (**b**) emerge only in the thermodynamic limit $N \to \infty$. **c** Recently experimentally observed discontinuous synchronization in a finite ($N = 200$) system of photo-chemical Belousov-Zabotinsky reactions (inset), modeled via FitzHugh-Nagumo fast-slow oscillators (main panel), data reproduced from[18]. **d** Extreme synchronization transitions in finite-$N$ systems of complexified Kuramoto units, visible already for $N = 8$. The inset displays $r$ vs. system size $N$ just above the critical coupling at $K = 1.05K_c$, in log-log scales with red dots for panel (**a**), blue dots for panel (**b**), and purple dots for panel (**d**). See Supplementary Information for details of the parameter settings.

Kuramoto oscillators with frequencies drawn from a bimodal distribution typically exhibit a discontinuous phase transition and also a gradual change of order for finite-$N$ systems,[17] see (Fig. 1b).

Interestingly, recent experiments on photo-chemically coupled relaxation oscillators that resemble Belousov-Zabotinsky (BZ) oscillatory reactions[18] found a discontinuous transition despite unimodally distributed natural frequencies (Fig. 1c), in contrast to the continuous transition found for phase-only oscillator. Curiously, these transitions often appear extreme, with (close to) maximal order $r \approx 1$ just past the transition. They remain largely unexplained to date.

In this article, we pin down and conceptualize an unprecedented class of discontinuous transitions—extreme synchronization transitions—in the collective dynamics of coupled oscillators. It qualitatively resembles the experimentally found transition for coupled BZ reactions,[18] cf. (Fig. 1c). We also identify the core mechanism underlying the extreme nature of extreme transitions. To be able to analytically access the transition, we first analyze coupled complexified Kuramoto oscillators[19,20]

$$\frac{d}{dt}z_\mu = \omega_\mu + \frac{K}{N}\sum_{\nu=1}^{N}\sin(z_\nu - z_\mu),\qquad(2)$$

for $\mu \in [N]$ with complex variables $z_\mu = x_\mu + \mathrm{i}y_\mu \in \mathbb{C}$ and coupling strength $K = |K|e^{\mathrm{i}\alpha} \in \mathbb{C}$. The model (2) analytically continues coupled phase-only oscillators and offer mathematical access to systems of finite units[19,20]. The natural frequency of each oscillator is a constant $\omega_\mu \in \mathbb{R}$ randomly independently drawn from a Gaussian distribution $g(\omega) = \frac{1}{\sigma\sqrt{2\pi}}e^{-\frac{\omega^2}{2\sigma^2}}$ with $\sum_{\mu=1}^{N}\omega_\mu = 0$, thus considering a co-moving reference frame. We provide numerical evidence of similar extreme transitions in other systems towards the end of this article and in the Supplementary Information.

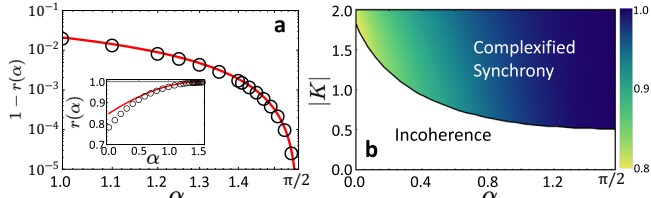

**Fig. 2 | Extreme features of the synchronization transition. a** The gap $1-r$ of the order parameter $r$ to its maximal value 1 for fixed $|K| = 1.5$. Direct numerical observations (open circles) agree well with our approximate, second-order prediction (8) asymptotically as $\beta = \pi/2 - \alpha \to 0^+$ (red line) and even for the full range of $\alpha \in (0, \pi/2)$ (inset). **b** With increasing $\alpha$, the critical coupling strength decreases and the jumps in order parameter $r$ (color-coded $r \in [0.8, 1]$) become increasingly extreme. The white area indicates an incoherent state with $r$ of the order of $N^{-1/2}$. The black solid curve indicates the critical coupling $|K_c|$. In (**a**) and (**b**), $N = 80$ across all observations.

## Results

### Observations: Finite-$N$ bifurcations with an extreme jump in the order parameter

Similar to the coupled BZ oscillators, we find a discontinuous transition (Fig. 1d) that does not require a thermodynamic limit and already emerges for systems with as few as $N = 8$ units (Fig. 1d, inset). It thereby constitutes a bifurcation in multi-dimensional systems.

Importantly, the transition is extreme in the sense that for small

$$\beta = \pi/2 - \alpha \in \left[0, \frac{\pi}{2}\right] \qquad (3)$$

the phase order parameter jumps from low values with $r = \mathcal{O}(N^{-1/2})$ in the incoherent state to values close to its maximum at $r = 1$ just past the transition point marked by a defined critical coupling strength, as Fig. 2 illustrates. The jump in order parameter is already extreme at moderate $\beta$. For instance, for $\beta < 0.4$, we already find $r > 0.99$ immediately past the transition. As $\beta$ approaches 0, the gap $1-r$ becomes arbitrarily small and thus the order parameter $r$ arbitrarily close to unity, already at moderate or even small coupling strength $|K|$, see also Fig. 2.

### Asymptotic analysis confirms extremeness

At $\beta = 0$ (and thus $\alpha = \frac{\pi}{2}$) and any given $|K| > 0$, the dynamical system (2) exhibits a fixed point

$$z_\mu^{(0)} = x_\mu^{(0)} + \mathrm{i}y_\mu^{(0)} = -\mathrm{i}\sinh^{-1}\left(\frac{b\omega_\mu}{|K|}\right),\qquad(4)$$

a complex locked state with identical $x_\mu^{(0)}$ for all $\mu \in [N]$. It is noteworthy that we numerically observe other fixed point solutions that are unstable and irrelevant to synchronization phenomena. To obtain Eq. (4), we initially assume the imaginary parts to be of the form $y_\mu^{(0)} = -\sinh^{-1}\left(\frac{b\omega_\mu}{|K|}\right)$ based on numerical observations and later confirm this form analytically (see Supplementary Information). From ansatz (4) and the fixed point conditions for both $x_\mu$ and $y_\mu$, we obtain a self-consistency condition

$$\frac{1}{b} = \frac{1}{N}\sum_{\nu=1}^{N}\sqrt{1 + \frac{b^2\omega_\nu^2}{|K|^2}}\qquad(5)$$

for the parameter $b > 0$. Moreover, a linear stability analysis together with a systematic numerical nonlinear analysis demonstrates that the fixed point is neutrally stable.

Interestingly, these complex locked states (4) exhibit identical synchronization with completely homogeneous phase variables,

$x_\mu^{(0)} = 0$ for all $\mu$ and thus $r = 1$, despite the heterogeneous natural frequencies directly driving the variables $x_\mu$ and the coupling strength $|K|$ being finite (and possibly small or moderate). We analytically quantify this extreme form of synchrony also for $\beta > 0$. We derive an asymptotic expansion[21,22] of the form $z_\mu^* = x_\mu^* + iy_\mu^* \sim x_\mu^{(0)} + x_\mu^{(1)}\beta + i(y_\mu^{(0)} + y_\mu^{(1)}\beta)$ as $\beta \to 0^+$ with $x_\mu^{(0)} = 0$ and $y_\mu^{(0)} = -\sinh^{-1}\left(\frac{b\omega_\mu}{|K|}\right)$ and real first order coefficients $x_\mu^{(1)}$ and $y_\mu^{(1)}$ for all $\mu$. Substituting this ansatz into (2) yields

$$
\begin{aligned}
z_\mu^* &\sim -Q\beta\tanh(y_\mu^{(0)}) + iy_\mu^{(0)} \\
&= Q\beta\frac{b\omega_\mu/|K|}{\sqrt{1 + \left(\frac{b\omega_\mu}{|K|}\right)^2}} - i\sinh^{-1}\left(\frac{b\omega_\mu}{|K|}\right)
\end{aligned}
\tag{6}
$$

up to corrections of order $\mathcal{O}(\beta^2)$ as $\beta \to 0^+$. Here

$$
Q = \frac{\sum_{\nu=1}^{N}\cosh y_\nu^{(0)}}{\sum_{\nu=1}^{N}\cosh y_\nu^{(0)} + \sum_{\nu=1}^{N}\sinh y_\nu^{(0)}\tanh y_\nu^{(0)}} > 0
\tag{7}
$$

is a positive parameter. For small $\beta > 0$, this first-order asymptotic result well characterizes the complex locked states, see Supplementary Information for an illustration (Figs. S1 and S2). With the relation (6), the order parameter (1) becomes

$$
\begin{aligned}
r(\beta) &= \frac{1}{N}\sum_{\nu=1}^{N}\cos(-Q\beta\tanh y_\nu^{(0)} + \mathcal{O}(\beta^2)) \\
&= 1 - \frac{1}{2}W_2 Q^2\beta^2 + \mathcal{O}(\beta^4)
\end{aligned}
\tag{8}
$$

for $0 \le \beta \ll 1$ (Fig. 2a). We remark that at least up to $\mathcal{O}(\beta^4)$, the imaginary part components of the complex locked state alone determine the constant $W_2 := \frac{1}{N}\sum_{\mu=1}^{N}\tanh^2(y_\mu^{(0)})$ as well as $Q$, because the real parts are $x_\nu^{(0)} = 0$. Moreover, the constants $W_2$ and $Q$ are essentially independent of the system size $N$ (see Supplementary Information; Fig. S3), resulting in the extremeness of synchronization already for small, finite $N$ (compare Fig. 1d inset). The relation (8) thus confirms analytically that the jump in order is extreme, with the difference $1 - r(\beta)$ decaying to zero quadratically, as $\beta \to 0^+$ (Fig. 2a). The order parameter hence exhibits an extreme jump from incoherence, where $r \propto N^{-1/2}$ to values close to maximal coherence where $r = 1 - \mathcal{O}(\beta^2)$ and thus close to unity for a range of $\beta$ (Fig. 2b) and essentially independent of $N$.

## How does such an extreme transition emerge?
First, how many heterogeneous parameters (here: the natural frequencies) that drive the rate of change of the phase variables $x_\mu$ yield collective states with these variables being close to identical and thus highly homogeneous? We find that whereas the order in the $x_\mu$ jumps extremely, with the order parameter close to unity just past the transition, the order in the other variables $y_\mu$ continuously and slowly grows with $|K|$, absorbing the disorder among the units (Fig. 3a). We furthermore qualitatively find that with varying $\beta$ and thus varying $\alpha = \pi/2 - \beta$ the heterogeneity in the state variables gradually transfers from the $x_\mu$ (at $\alpha = 0$) to the $y_\mu$ (at $\alpha \to \pi/2$) in multi-dimensional state space (Fig. 3b). Moreover, a numerical quantitative analysis shows that the local angle $\varphi = \arg z_\mu$ and thus $\tan\varphi = \frac{y_\mu}{x_\mu}$, computed for sufficiently small $|x_\mu|$ and $|y_\mu|$, well matches the negative argument of the complex parameter $K$, i.e. $\varphi \approx -\alpha$, see Fig. 3c. An asymptotic analysis (see Supplementary Information; Fig. S4) for the simplest coupled system of $N = 2$ units confirms this finding as it yields

$$
\tan\varphi = -\tan\alpha + \mathcal{O}\left(\left|\frac{\omega_2 - \omega_1}{K}\right|^2\right),
\tag{9}
$$

indicating that the one-to-one argument mapping between parameters and collective states may hold more and more exactly as $|K|$ grows. Overall, the parameter disorder driving one set of variables, the $x_\mu$, is gradually redistributed or transferred to other variables of the system, here the $y_\mu$.

Second, given the transition occurs already for finite $N$, it constitutes a bifurcation of a finite-dimensional dynamical system, not a phase transition that requires a thermodynamic limit $N \to \infty$. Analyzing the eigenvalue spectrum of the (complex) locked state, we find that one pair of eigenvalues crosses the imaginary axes from positive to negative real parts upon increasing $|K|$, signifying a Hopf bifurcation, see Fig. 4a–c. Numerical evaluations of the order parameter $r$ as a function of time Fig. 4d–f support this view. Indeed, oscillatory dynamics of $r(t)$ with the characteristic time scale are already apparent below the transition point, panel Fig. 4d.

## Extreme synchronization transitions emerge across different systems
The current work has been motivated by experimental observations of such transitions in coupled Belousov-Zhabotinsky chemical reactions[18] that may be modeled by coupled Fitzhugh-Nagumo oscillators (Fig. 1c). Our study of the complexified Kuramoto model with all-to-all coupling has enabled analytic access to collective states representing synchrony (complex locked states) as well as the order parameter. Further systematic numerical simulations indicate that features of extreme transitions may also exist in other systems of coupled oscillators, including in systems without all-to-all coupling and networks with random interaction topologies (see Supplementary Information; Fig. S5), in coupled van-der-Pol oscillators that represent a class of coupled relaxation oscillators as well as in Stuart-Landau oscillators that represent a class of phase-amplitude oscillators (Fig. 5).

In general, the extreme transitions we report thus come with four characteristics:

(i) extremeness: discontinuous jump to near-maximal ordering just above a defined critical coupling,
(ii) finite-size systems: emergence already for finite systems, possibly of moderate or small sizes; no requirement of a thermodynamic limit $N \to \infty$,
(iii) transition at low $K_c$ (in contrast to at large $K_c$ induced by delayed transitions as in certain explosive transitions), and
(iv) redistribution of parameter disorder to variables other than those driven by the heterogeneous parameters

Condition (i) is a requirement for justifying the transition to be called *extreme*, yet the other features (ii)-(iv) may or may not co-occur in a given order-disorder transition. It is conceivable, for instance, that coupled van der Pol oscillators may exhibit a delayed synchronization transition at large $K$ [thus not exhibiting characteristic (iii)]. If so, coupling strengths slightly above the critical value may overcome parameter heterogeneities by far, as for the original Kuramoto model for very large $K$, removing the need to transfer disorder to other variables (iv). Similarly, Stuart-Landau oscillators exhibit an extreme synchronization transition (Fig. 5b) for large $K$ by quenched oscillation,[23] i.e., a fixed point solution. However, during the synchronization transitions, the redistribution of parameter disorder into amplitude degrees of freedom is not readily apparent.

## Distinct nature of the transition
The nature of such extreme transitions stands in contrast to standard synchronization transitions found in the paradigmatic Kuramoto model, with continuous phase transitions for unimodal natural frequencies[15] and with discontinuous phase transitions for bimodal[17] (or bounded-support[24]) natural frequency distributions, see also[7,9]. Both emerge only in the thermodynamic limit $N \to \infty$ and exhibit moderate order, perhaps $r \approx 0.7$ for moderate $K > K_c$ past the transition.

Intriguingly, our asymptotic analysis (6) indicates that the extreme transitions found here emerge independently of the specific natural frequency realizations, examples ranging from uniform, unimodal, bimodal and even to tri-modal distributions (see Supplementary Information; Fig. S6 and Fig. S7), in contrast to the original Kuramoto model where the class of phase transition depends on the form of the

frequency distributions (for instance, unimodal vs. bimodal,[17] and bounded vs. unbounded support[24]).

The extreme transition we found emerges already for small system sizes $N$ with, for instance, $r > 0.99$ for $N = 8$ coupled units immediately past $|K| > K_c$, compare Fig. 1d, inset. We remark that some oscillator systems may exhibit discontinuous synchronization transitions even for a few coupled units, yet typically, these are not extreme transitions nor analytically accessible. In contrast, our study offers enhanced analytical access, even for finite-sized systems, and pins down the extreme nature of synchronization transitions as well as the behavior of the order parameter. Moreover, the transition emerges at coupling strength $K_c$ substantially smaller than that of the original Kuramoto model, becoming even smaller as $\alpha$ increases. Furthermore, this earlier transition stands in stark contrast to known, explosive phase transitions[7,25–31] that have been identified in both temporal and structural ordering processes, specifically for synchronization and percolation, where the transitions are often delayed to larger $K_c$, that in turn contributes to a strong jump in order parameter once the transition occurs. Intriguingly, recent studies[32–34] observed a percolation transition that appears to also exhibit extreme features in the sense we introduced above.

Previously known discontinuous (non-extreme) transitions, including in systems with higher-order interactions (e.g., biharmonic interactions[35]), occur through a bistability between incoherent and synchronous states[7]. Such bistability arises due to a change from a supercritical to a subcritical branching point bifurcation (e.g., a pitchfork or transcritical bifurcation). For continuous transitions, stable synchrony bifurcates directly off the incoherent state at a supercritical branching point. For discontinuous transitions, a branch of unstable synchrony connects the incoherent state and with the stable synchronous state. In contrast, further analysis of the extreme synchronization transitions we report indicates that the stable (extremely) synchronous state is disconnected from the incoherent state.

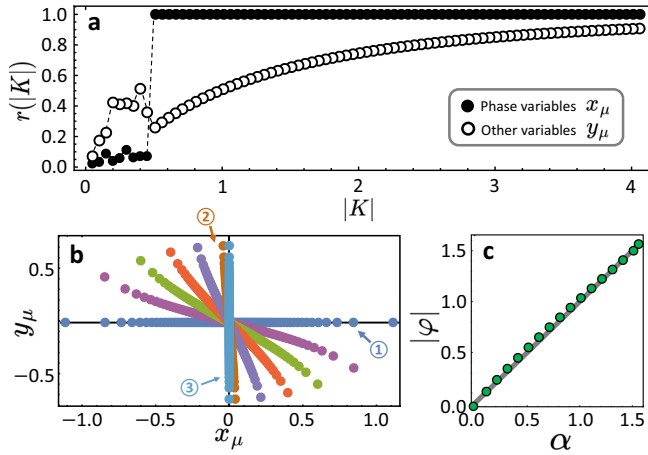

**Fig. 3 | Disorder moves to additional variables. a** The order parameter $r$ is depicted as a function of $|K|$ for both real parts, i.e. the phase-variables (solid disk) and the other variables (the imaginary parts, open circles) for $N = 128$ and $\beta = 0.01$. As the $y_\nu$ are unbounded, we define phase-like variables $\theta_\mu$ by a stereographic projection via $\cos\theta_\mu := \frac{1-y_\mu^2}{1+y_\mu^2}$ and $\sin\theta_\mu := \frac{2y_\mu}{1+y_\mu^2}$ for each $\mu$ and evaluate $r = |\frac{1}{N}\sum_{\mu=1}^{N} e^{i\theta_\mu}|$, in analogy to (1). **b** Complex locked states in the complex plane for $N = 80$ and $|K| = 3.0$ move with increasing $\alpha$ values from curves ① for $\alpha = 0$ and ② for $\alpha = 1.5$ to curve ③ for $\alpha = \frac{\pi}{2}$. **c** Local angles $\varphi$ of the curves around the origin are depicted as a function of $\alpha$ with gray solid guiding line indicating $|\varphi(\alpha)| = \alpha$ as emerges for $N = 2$ up to corrections $\mathcal{O}(|K|^{-1})$.

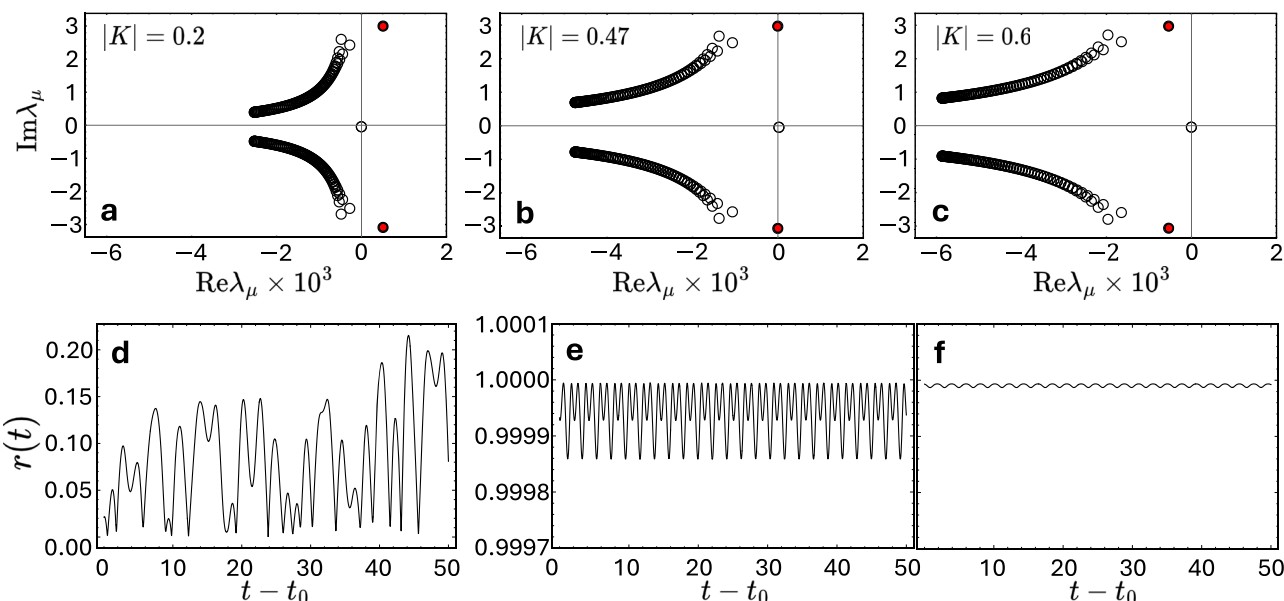

**Fig. 4 | Extreme synchronization emerges via Hopf bifurcation.** Panels (**a**–**c**) display the eigenvalues (open circles) of the Jacobian matrix evaluated at the locked state $z^*$. The pair of eigenvalues relevant to the bifurcation is highlighted by filled red disks. It crosses the imaginary axis with increasing $|K|$, indicating a Hopf bifurcation. For panel (**b**), we choose $|K| = 0.47$, close to but slightly above the critical coupling strength. **d**–**f** show the order parameter as a function of time after

a transient period, $t_0 = 3000$, with the system state initiated by a random perturbation of order $10^{-1}$ away from each locked state evaluated in (**a**–**c**), respectively. Additional oscillations visible in (**e**) and (**f**) are transient phenomena due to small negative real parts of eigenvalues. All panels for $\alpha = \frac{\pi}{2} - 0.01$, i.e., $\beta = 0.01$ and $N = 128$.

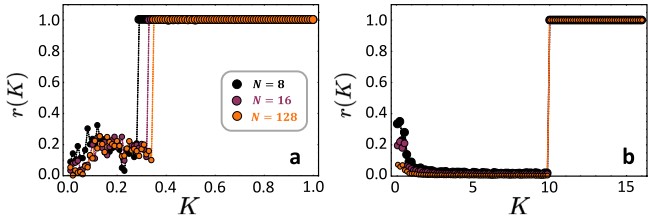

**Fig. 5 | Extreme transitions in other systems. a** Systems of relaxation (Van der Pol) oscillators and (**b**) of phase-amplitude (Stuart-Landau) oscillators with Gaussian natural frequency distribution, both exhibit discontinuous transitions in finite systems with an order parameter close to its maximum immediately past the transition point. Exact governing equations, parameters, and a definition of phases entering the order parameter $r$ in (1) are detailed in the Supplementary Information.

Interestingly, recent experiments[18] on photochemically coupled Belousov-Zabotinsky oscillators already hinted at similar extreme transitions (cf. Fig. 1c). The authors in[18] raised a hypothesis that a sudden synchronization transition they observed emerges "beyond phase reduction," yet did not identify the detailed mechanism of the extreme nature of their observed synchronization transitions. It is noteworthy that a model representing the Belousov-Zhabotinsky reaction consists of planar oscillators beyond standard phase reduction. Such traditional planar oscillator models often fail to elucidate the core mechanism of the onset of extreme synchronization. Our results confirm that more than just phase state variables are necessary and emphasize the potential finite-$N$ and extreme nature of the transition.

## Discussion

We have presented and analyzed an unprecedented form of transition, an extreme synchronization transition, that constitutes an intriguing instance of a bifurcation in finite, multi-unit nonlinear dynamical systems.

More generally, these results offer an alternative perspective for interpreting the extreme nature of certain explosive phenomena, encouraging to rethink of the underlying mechanisms behind the onset of explosive synchronization transitions. For example, coupling adaptivity[36] might absorb parameter heterogeneities in oscillator frequencies, which would otherwise prevent synchrony such that the transition becomes discontinuous and may potentially become extreme. More broadly, higher-dimensional oscillator models may effectively absorb parameter disorder in additional variables[37].

From a theoretical and methodological perspective, analytically continuing variables and parameters to become complex has previously advanced our understanding of fractals,[38] phase transitions in statistical physics,[39,40] and the foundations of quantum mechanics[41–44]. Our analytic continuation of coupled oscillator systems has revealed an unprecedented class of synchronization transition and clarified the core mechanisms underlying it, underlining that analytic continuation may also be valuable in understanding emergent properties of networked nonlinear dynamical systems[45].

The extreme synchronization transition studied above may have general relevance to real-world problems. In applications, the occurrence of extreme transitions impacts our ability of ensuring or preventing strong forms of synchrony. For instance, strong synchrony shall be avoided in neural diseases such as Parkinson's or epilepsy, yet extreme transitions may induce strong synchrony immediately past seizure onset. Similarly, while synchrony in terms of phase locking is required to operate an electric power grid, strong synchrony in the sense of close-to-identical phases prevents efficient power flow between network nodes[46]. We may exploit extreme transitions in technical systems such as swarmalators, compare[47]. For example, when implementing coupled nonlinear oscillator systems with

extreme transition features in swarms of unmanned aerial vehicles (UAVs) or ground-based robots, we may exploit the particular variables that exhibit extreme synchronization transitions to enable a tightly synchronized state that in turn enables a self-organized clocking scheme for robust communication[48–50]. How and under which conditions extreme transitions to synchrony may arise in such different systems also constitute open questions for future research.

Our results raise a number of intriguing conceptual and technical questions. For instance, which role do the sizes of the basins of attraction and hysteresis phenomena play in making transitions extreme or in inducing such pronounced transitions already at small or moderate system size (see Supplementary Information for examples: Fig. S8)? Under which conditions does intrinsic disorder shift to other variables or, more generally, does each of the features (i)-(iv) that indicate an extreme transition also arise beyond synchronization phenomena?

## Data availability

The data can be reproduced from the codes in *Mathematica*, which are publicly accessible: https://github.com/NetworkSync/ExtremSyncTrans.git. The data that support the findings of this study are also available from the corresponding author upon request.

## Code availability

We did not use any specific custom-made code; our analyses relied on standard ODE solvers, algebraic equation solvers (root finders) and eigenvalue solvers available in *Mathematica*, which are publicly accessible: https://github.com/NetworkSync/ExtremSyncTrans.git.

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

## Acknowledgements

We thank Moritz Thümler and Malte Schröder for providing a fruitful discussion about the dynamics of complexified dynamical systems. This work has been partially supported by German Federal Ministry for Education and Research (BMBF) under grant number 03SF0769 (ResiNet) as well as the German National Science Foundation (Deutsche Forschungsgemeinschaft DFG) under grant number DFG/TI 629/13-1 (SynCON) and under Germany's Excellence Strategy - EXC-2068 - 390729961, through the Cluster of Excellence *Physics of Life*.

## Author contributions

M.T. conceived the research. S.L. and M.T. designed research. S.L. performed theoretical analysis, supported by M.T., and numerical simulations, supported by L.J.K. All authors contributed to the interpretation of theoretical results and numerical data. S.L., supported by L.J.K. and M.T., created the Figures. M.T. supervised the work. S.L. and M.T. wrote the first draft and edited the article.

## Funding

## Competing interests

The authors declare no competing interests.
