## [Transparent Peer Review file · Nature Communications]

Extreme Synchronization Transitions

Corresponding Author: Dr Seungjae Lee

Version 0:

Reviewer comments:

Reviewer #1

(Remarks to the Author)

SUMMARY

The article "Extreme Synchronization Transitions" explores a novel class of transitions in coupled oscillators, termed extreme synchronization transitions. The extreme transitions involve a sudden shift from disordered to highly ordered states, resembling explosive phase transitions but with unique characteristics. Unlike traditional phase transitions that require an infinite number of units, these transitions occur in finite systems and involve a significant jump in the synchronization order parameter to near-maximum values immediately after crossing a critical coupling strength. The study uses complexified Kuramoto oscillators to analytically explain these transitions and suggests that similar phenomena could occur in other systems of coupled oscillators. Indeed, temporal ordering of oscillators in the Kuramoto phase oscillator model can be seen as temporal analogy of transitions such as the emergence of ferromagnetism or freezing in many-particle systems. The findings have implications for understanding and controlling synchronization in both biological and engineered systems, highlighting the potential for extreme transitions to impact the predictability and management of system behaviors. Furthermore, the study reveals that the extreme transition from incoherence to (nearly) perfect coherent oscillations is possible even for non-identical oscillators and low numbers of oscillators, in which, classically, one would expect a slower continuous onset of synchronization due to the heterogeneity. The study also confirms this phenomenon in an asymptotic analysis, and explains how the heterogeneity is absorbed into the secondary imaginary variable, which then allows for a sudden onset of synchrony in real variables -- a very interesting finding.

The paper is very well written and accessible to a broad audience, the topic is timely and of general interest. The results on the extreme transition are very interesting, and proposes a classification scheme that will guide future studies. The submitted manuscript is therefore suitable for a publication in Nature Communications.

ASSESSMENT: I strongly recommend publication of the manuscript, provided that comments or concerns by the reviewers can be addressed satisfactorily.

COMMENTS:

I do not see any problems in the methodology/analysis of the paper, it is carried out very carefully.

I have some general comments of more general, scientific nature:

1. Sudden synchronization transitions may occur in systems with biharmonic coupling (e.g., Kuramoto model). Such systems exhibit bistability between incoherence and synchrony and thus sudden transitions, even for low oscillator numbers. I am not sure how this looks for heterogeneous oscillators, but it seems reasonable to also consider and compare with this case for a deeper (and more general) discussion. Some readers may think of biharmonic coupling as a natural mechanism for a sudden onset of synchrony.

Reference (e.g., there are others and perhaps more suitable references):

León, I., Muolo, R., Hata, S., & Nakao, H. (2024). Higher-order interactions induce anomalous transitions to synchrony. *Chaos: An Interdisciplinary Journal of Nonlinear Science*, 34(1).

2. Parallels are drawn to several other oscillator systems:

i) [13] D. Călugăru, et al. In this study the phase response curve is reported, displaying a shape with a sharp saw-tooth like bump. This indicates that a phase model involves a coupling function (interaction between phases) represented by a Fourier series with higher order harmonics. Biharmonic (or higher order harmonic interactions) are now known to exhibit bi-stable states (see point 1.), and thus may lead to a sudden (or, extreme) onset of synchronization. It could be a likely alternative explanation for the sudden onset of synchronization, but I do not want to rule out the explanation presented in the paper. Yet, which variable would absorb the heterogeneity in the BZ oscillator model, as is the case of the complex KM?

ii) System of coupled van der Pol oscillators: A single forced van der Pol oscillator goes through a Hopf bifurcation generating tiny oscillation amplitude before undergoing a Canard explosion (it is relatively easy to see while using logarithmic increments for the forcing strength). Couldn't that be a likely alternative explanation for the sudden onset of synchrony, as presented in Fig. 5?

iii) System of coupled Stuart-Landau oscillators (or, complex Ginzburg-Landau oscillators): the Stuart-Landau model is known to exhibit amplitude death (aka quenching). Couldn't that be a likely alternative explanation for the sudden onset of synchrony, as presented in Fig. 5 (else, why not)?

References (e.g.):

Koseska, A., Volkov, E., & Kurths, J. (2013). Oscillation quenching mechanisms: Amplitude vs. oscillation death. *Physics Reports*, 531(4), 173-199.

Chabanol, M. L., Hakim, V., & Rappel, W. J. (1997). Collective chaos and noise in the globally coupled complex Ginzburg-Landau equation. *Physica D: Nonlinear Phenomena*, 103(1-4), 273-293.

3. Finally, it may be interesting to note that systems with adaptive coupling and heterogeneous oscillator frequencies experience a widening of the synchronization region as the adaptive capability of the system is increased (effectively widening the associated Arnol'd tongue). It would appear as if the adaptive coupling may be able to absorb the effect of heterogeneity in oscillator frequencies, which would destroy synchrony in the absence of adaptive couplings. More generally, it would appear that oscillator models that effectively are higher dimensional, as is the case with the complexified Kuramoto model, have the ability to absorb heterogeneities in other variables.

References (e.g.):

Ren, Q., & Zhao, J. (2007). Adaptive coupling and enhanced synchronization in coupled phase oscillators. *Phys. Rev. E* 76(1), 016207.

MINOR COMMENTS:

- The notation clash for g (in Eq. 7) and the oscillator frequency, $g(\omega)$, clash.

- Why not use either α OR β as axis in Fig 2? They effectively show the same (I was at first confused and looking for incommensurate data types in the figure).

- The time evolution of the complex order parameter, $Z(t)$, in the complex plane can be more revealing to see features. E.g., for the classical Kuramoto model, for time going to infinity, in the incoherent regime where $K < K_c$, the trajectory will be arbitrarily close to the origin, while it will never get close to the origin once $K < K_c$. Thus, there is a clear onset of synchrony visible when reporting $\min_t(|Z(t)|)$.

Perhaps it is interesting to look at the oscillations reported in Fig .4d) in such a way.

Reviewer #2

(Remarks to the Author)

What are the noteworthy results?

See attached pdf.

Will the work be of significance to the field and related fields? How does it compare to the established literature? If the work is not original, please provide relevant references.

See attached pdf.

Does the work support the conclusions and claims, or is additional evidence needed?

The work supports its claims.

Are there any flaws in the data analysis, interpretation and conclusions? Do these prohibit publication or require revision?

No.

Is the methodology sound? Does the work meet the expected standards in your field?

Yes.

Is there enough detail provided in the methods for the work to be reproduced?

Yes.

Reviewer #3

(Remarks to the Author)

Reviewer #4

(Remarks to the Author)

Comments on 'Extreme Synchronization Transitions' by Seungjae Lee, Lennart J. Kuklinski and Marc Timme.

The manuscript and Supplementary Material "Extreme Synchronization Transitions" present an intriguing study of a class of discontinuous collective dynamics in coupled oscillator systems. The authors explore extreme synchronization transitions, characterized by an abrupt leap from disordered states to near-complete synchrony, even in finite systems. Motivated by experimental observations in coupled Belousov-Zhabotinsky reactions, the study systematically investigates the dynamics and mechanisms underlying these transitions using coupled complexified Kuramoto oscillators as a model system.

The work is notable for uncovering distinct features of extreme transitions: dramatic changes in the order parameter, independence from thermodynamic limits, and a redistribution of parameter disorder among system variables. These findings are relevant to fundamental studies in nonlinear dynamics and practical applications in biological and engineered systems, including mitigating pathological synchrony in neural disorders or optimizing synchronization in power grids and robotic swarms.

I believe this manuscript is well-suited for publication in Nature Communications following revision. Below, I provide a detailed list of questions and suggestions to enhance the clarity and impact of the study.

Main Questions:

1. Clarification on the Complexified Kuramoto Model: The Kuramoto model is widely recognized as a phase model derived from amplitude systems, with analogs in real-world applications. However, the concept of the complexified Kuramoto model remains somewhat unclear. Beyond the synchronization phenomena observed in Belousov-Zhabotinsky reactions, are there other real-world systems or scenarios where such a model has direct relevance? Providing concrete analogs or applications would help contextualize its significance. For instance, the manuscript provides examples of systems where extreme transitions might impact synchrony, such as neural diseases (e.g., Parkinson's or epilepsy) and power grids. However, it is unclear how the specific dynamics of the complexified Kuramoto model connect to these examples. Could the authors clarify whether the model presented in this study has direct applicability to these systems? If these examples are meant to illustrate general relevance rather than a direct connection, explicitly stating this would help set appropriate expectations for readers.
2. Thermodynamic Limit and Finite System Size: The authors emphasize that extreme synchronization transitions occur in finite systems, without requiring the thermodynamic limit. However, not all discontinuous synchronization transitions rely on the thermodynamic limit. For example, models with bimodal frequencies, Janus oscillators, or setups for explosive synchronization can exhibit similar phenomena without such requirements. Given these precedents, what is the specific theoretical or practical importance of emphasizing this distinction in the context of extreme synchronization? This clarification would strengthen the manuscript's conceptual contribution.

Minor Questions:

1. Presentation of the Kuramoto Model: In the manuscript, the authors discuss the Kuramoto model as a foundational concept before introducing the complexified Kuramoto model. While this sets up the context, the figures illustrating results (e.g., those based on coupling strength) appear before the governing equations and definitions are introduced. This presentation could be confusing for readers unfamiliar with the Kuramoto model or its modifications. Could the authors consider including the equations and key definitions earlier in the text, ideally before presenting the simulation results?
2. Definition of the Phase-Like State Variable: In calculating the order parameter, the manuscript references the state vector $x_{\nu} \in \mathbb{R}$. However, it is not entirely clear what is meant by the "phase-like" state variable mentioned in the manuscript. If x_{ν} is not the conventional phase variable used in the Kuramoto framework, could the authors clarify its meaning and how it relates to the phases typically used in synchronization studies? Providing a concise definition of x_{ν} and how it differs from or extends conventional phase variables would enhance clarity. This question might also be relevant to the broader question about the complexified Kuramoto model (see Main Question 1).
3. Legend Transparency in Figure 1: In Figure 1, the legend uses varying transparency levels to distinguish system sizes, with transparency increasing from large systems to small ones. This choice is not immediately intuitive and may be confusing for readers. Could the authors clarify the rationale behind this design choice? Alternatively, adopting a more conventional style, such as distinct colors or line styles, might improve readability.
4. Color Discrepancy in Figure 3: The caption of Figure 3 refers to phase and other variables using purple and orange colors, respectively. However, these colors do not appear in the plot, which may lead to confusion. Could the authors verify and correct the caption or the figure itself to ensure consistency?

The manuscript provides significant insights into extreme synchronization transitions and their implications across various systems. The theoretical framework is rigorous, and the supplementary materials offer an excellent resource for readers interested in additional technical details. However, addressing the questions above, particularly those related to the real-world relevance of the complexified Kuramoto model, would further enhance the clarity and impact of the work.

I recommend publication following these revisions, as the findings are likely to attract broad interest within the readership of Nature Communications.

Version 1:

Reviewer comments:

Reviewer #2

(Remarks to the Author)

The authors have addressed all of my comments. I am now happy for this article to be published.

Reviewer #3

(Remarks to the Author)

Reviewer #4

(Remarks to the Author)

The authors have provided clear, well-reasoned responses to our and the other reviewers' comments. They have adequately addressed the raised concerns, and their decision to maintain the current structure for certain recommendations is justified by their comprehensive answers. I believe the manuscript has been significantly improved, and I am happy to support its publication in Nature Communications.

Replies to the comments of the Reviewers

on the manuscript

Extreme Synchronization Transitions

by Seungjae Lee, Lennart J. Kuklinski, and Marc Timme

GENERAL REPLIES

We thank all Reviewers for carefully assessing our manuscript and for their positive feedback and detailed comments. Their valuable comments and suggestions have helped us further improve the manuscript.

In the comments, the Reviewers noted that our study is accessible to a broad audience, the topic is timely and of general interest, the findings have implications for understanding and controlling synchronization in complex systems, highlighting the potential for extreme transitions to impact the predictability and management of system behaviors. Moreover, the Reviewers emphasized that the general findings and discussion on the extreme transitions are interesting and raise important questions about how we should view explosive synchrony transitions in systems with partial observable subsets of their variables. Furthermore, two Reviewers remarked that the work is notable for uncovering distinct features of extreme transitions and is relevant to fundamental studies in nonlinear dynamics, with practical applications in biological and engineered systems.

Overall, the Reviewers commented that the manuscript provides significant insights into extreme synchronization transitions and their implications across various systems, supported by a rigorous theoretical framework and supplementary materials that serve as an excellent resource for readers seeking additional technical details. These arguments mainly support the publication of our work in *Nature Communications*.

The Reviewers also have a number of comments, critical questions and suggestions for improving the manuscript and asking to delineate our results from existing concepts, in particular our own recent publications and the phenomenon of explosive transitions. We address all of those in our point-by-point replies below.

Given some of the critiques and questions, let us take the opportunity to briefly summarize the core novel contributions made in the current manuscript.

- (i) Extremeness: We have discovered and pinned down a novel feature in discontinuous (explosive) transitions, where the order parameter jumps to a value extremely close to its maximum immediately past the critical point despite heterogeneous natural frequencies.
- (ii) Persistence in finite-size systems: The phenomenon does not rely on the thermodynamic limit but persists in finite systems of only a few coupled units.
- (iii) Continuation method enables analytic access: We have constructively used the complexified, analytically continued Kuramoto model to explicitly mathematically demonstrate the new properties of extreme transitions and the order parameter in finite systems – an endeavor typically impossible for most systems.
- (iv) Core mechanism: We have revealed that and quantified how an absorption of intrinsic parameter disorder into additional degrees of freedom enables the extreme nature of such transitions.

In particular, our study qualitatively expands the class of mechanisms behind the onset of explosive synchronization transitions, offering a novel perspective with broad applicability across synchronization transitions and other explosive phenomena.

To address all of the Reviewers' comments and feedback, we have thoroughly revised the manuscript. In particular, we have: (i) enhanced the explanation of how a novel extreme synchronization transition studied in the current article goes beyond widely known discontinuous transitions to synchrony, e.g. for higher-order interactions and for planar oscillator systems, as requested mainly by Reviewer #1; (ii) emphasized the novelty of the findings

we report in the current study compared to prior works, and discussed what distinguishes our study on extreme synchronization transitions from commonly studied explosive transition phenomena, as highlighted mainly by Reviewer #2; and (iii) clarified the complexified Kuramoto model and the applicational perspectives of the result, as pointed out mainly by Reviewer #4. Moreover, we modified the manuscript to incorporate aspects based on comments provided by the Reviewers, also clearly highlighting all the messages in the manuscript to a broad potential readership of *Nature Communications*.

In the revised manuscript, we now more succinctly emphasize the importance of the topic and novel findings as well as the core mechanism underlying extreme transitions. Below, we provide detailed point-by-point responses to each individual comment.

REPLIES TO THE COMMENTS OF REVIEWER #1

Reviewer comment:

SUMMARY

The article "Extreme Synchronization Transitions" explores a novel class of transitions in coupled oscillators, termed extreme synchronization transitions. The extreme transitions involve a sudden shift from disordered to highly ordered states, resembling explosive phase transitions but with unique characteristics. Unlike traditional phase transitions that require an infinite number of units, these transitions occur in finite systems and involve a significant jump in the synchronization order parameter to near-maximum values immediately after crossing a critical coupling strength. The study uses complexified Kuramoto oscillators to analytically explain these transitions and suggests that similar phenomena could occur in other systems of coupled oscillators. Indeed, temporal ordering of oscillators in the Kuramoto phase oscillator model can be seen as temporal analogy of transitions such as the emergence of ferromagnetism or freezing in many-particle systems. The findings have implications for understanding and controlling synchronization in both biological and engineered systems, highlighting the potential for extreme transitions to impact the predictability and management of system behaviors. Furthermore, the study reveals that the extreme transition from incoherence to (nearly) perfect coherent oscillations is possible even for non-identical oscillators and low numbers of oscillators, in which, classically, one would expect a slower continuous onset of synchronization due to the heterogeneity. The study also confirms this phenomenon in an asymptotic analysis, and explains how the heterogeneity is absorbed into the secondary imaginary variable, which then allows for a sudden onset of synchrony in real variables – a very interesting finding.

The paper is very well written and accessible to a broad audience, the topic is timely and of general interest. The results on the extreme transition are very interesting, and proposes a classification scheme that will guide future studies. The submitted manuscript is therefore suitable for a publication in Nature Communications.

ASSESSMENT: I strongly recommend publication of the manuscript, provided that comments or concerns by the reviewers can be addressed satisfactorily.

COMMENTS: I do not see any problems in the methodology/analysis of the paper, it is carried out very carefully.

Authors' response:

We thank the Reviewer for all the valuable comments and suggestions. We address the individual comments in the point-by-point replies below.

Reviewer comment:

I have some general comments of more general, scientific nature:

1. Sudden synchronization transitions may occur in systems with biharmonic coupling (e.g., Kuramoto model). Such systems exhibit bistability between incoherence and synchrony and thus sudden transitions, even for low oscillator numbers. I am not sure how this looks for heterogeneous oscillators, but it seems reasonable to also consider and compare with this case for a deeper (and more general) discussion. Some readers may think of biharmonic coupling as a natural mechanism for a sudden onset of synchrony.

*Reference (e.g., there are others and perhaps more suitable references): León, I., Muolo, R., Hata, S., and Nakao, H. (2024). Higher-order interactions induce anomalous transitions to synchrony. *Chaos: An Interdisciplinary Journal of Nonlinear Science*, 34(1).*

Authors' response:

We thank the Reviewer for this valuable and important comment that helps extend the scope of our findings. Inspired by the Reviewer's comment, we have now explicitly numerically tracked a transition for heterogeneous phase oscillators under higher-order interactions. Each oscillator satisfies

$$\frac{d}{dt}x_\mu = \omega_\mu + \frac{K_1}{N} \sum_{\nu=1}^N \sin(x_\nu - x_\mu) + \frac{K_2}{N^2} \sum_{\nu,\eta=1}^N \sin(2x_\nu - x_\eta - x_\mu) \quad (\text{R1})$$

for $\mu \in \{1, \dots, N\}$ where ω_μ is randomly drawn from a Gaussian distribution centered at zero with unit variance. In Fig. R1, the Kuramoto order parameter r is illustrated as a function of coupling strength K_1 . The result shows a discontinuous transition as expected (and known for related examples), but not an extreme transition (with order parameter close to unity just past the transition point). Crucially, it lacks the extreme nature of synchronization and the heterogeneity in natural frequencies induces a substantial remaining spread of the phases also past the transition point.

FIG. R1. **Discontinuous transitions for heterogeneous phase oscillators with higher-order interactions.** The Kuramoto order parameter r is depicted as a function of the coupling strength K_1 where $N = 128$ and $K_2 = 2$ with natural frequencies drawn from a Gaussian distribution of unit variance.

As noted in the comment, bistability between incoherence and synchrony serves traditionally as a natural mechanism for the onset of discontinuous or sudden synchronization. As we highlight further below, there are key differences in the core mechanisms between "*extreme synchronization transitions*" and traditional discontinuous or explosive transitions. In particular, the bifurcation type may differ.

In the revised manuscript, we provide the following explanation (Section "*Distinct nature of the transition.*") and provide a corresponding reference to an orthogonal study focusing on the persistence of discontinuous transitions more generally (S. Lee *et al.*, "*Persistently discontinuous phase transitions to synchrony*").

A traditionally known discontinuous transition, including for higher-order interactions (e.g., biharmonic interactions), occurs through bistability between incoherence and synchrony. This phenomenon arises in principle due to a change in criticality — from supercritical to subcritical — of a branching point bifurcation (e.g., a pitchfork or transcritical bifurcation). The details are discussed in C. Kuehn and C. Bick, *Science Advances* **7**, abe3824 (2021).

More precisely, for a continuous synchronization transition, stable synchrony bifurcates off the incoherent state at a *supercritical* branching point. On the other hand, a discontinuous transition is mediated by an unstable (low degrees of synchrony) branch that bridges the incoherent state and stable (high degrees of) synchrony. In contrast, for a discontinuous (or sudden, or explosive) transition, the unstable synchrony bifurcates off the incoherent state at the *subcritical* branching point and it becomes stable via a saddle-node bifurcation.

a. Traditional discontinuous transitions via a change of criticality

b. Extreme synchronization transitions

FIG. R2. **Fundamental difference between extreme synchronization transitions and traditional discontinuous transitions.** (a) Schematics for a traditional discontinuous transition that emerges via a change of criticality of a branching point bifurcation from supercritical (BP^+) to subcritical (BP^-). (b) Schematic illustration for an extreme synchronization transition. For both panels, solid curves indicate stable states and dashed curves indicate unstable states.

In line with illustrations in Fig. R2 (a) in this reply letter, the emergence of discontinuous synchronization in such (branching point) scenario is mediated by coupling strengths that are particularly large compared to parameter disorder, i.e. coupling strengths slightly above the critical value may overcome parameter heterogeneities by far, as pointed out on page 10 in the original manuscript (also by Kühn and Bick).

Such discontinuous transitions may exhibit extreme features in the order parameter, though not to the extent observed in our extreme synchronization transitions where the order parameter approaches values arbitrarily close to the maximal order, depending on the variation of a given system parameter, β .

In contrast, the "*extreme synchronization transitions*" studied in our manuscript (compare Fig. R2 (b)) follows a fundamentally distinct mechanism, the incoherent state is disconnected from the synchronous state, i.e. no branch exists that connects them. As a consequence, the branch representing synchrony for large coupling never reaches down to the zero value of the order parameter that characterizes the incoherent state. Moreover, the stabilization of unstable synchrony occurs via a Hopf bifurcation (see Fig.4 in the original manuscript) rather than a saddle-node bifurcation or branching point criticality. In this scenario, as revealed in our study, the extreme synchronization transition arises similarly for finite and even small systems. The redistribution of parameter disorder in the phase variables into additional degrees of freedom enables the system to achieve phase order close to its theoretical maximum.

Reviewer comment:

2. *Parallels are drawn to several other oscillator systems:*

i) [13] D. Călugăru, et al. In this study the phase response curve is reported, displaying a shape with a sharp saw-tooth like bump. This indicates that a phase model involves a coupling function (interaction between phases) represented by a Fourier series with higher order harmonics. Biharmonic (or higher order harmonic interactions) are now known to exhibit bi-stable states (see point 1.), and thus may lead to a sudden (or, extreme) onset of synchronization. It could be a likely alternative explanation for the sudden onset of synchronization, but I do not want to rule out the explanation presented in the paper. Yet, which variable would absorb the heterogeneity in the BZ oscillator model, as is the case of the complex KM?

ii) System of coupled van der Pol oscillators: A single forced van der Pol oscillator goes through a Hopf bifurcation generating tiny oscillation amplitude before undergoing a Canard explosion (it is relatively easy to see while using logarithmic increments for the forcing strength). Couldn't that be a likely alternative explanation for the sudden onset of synchrony, as presented in Fig. 5?

iii) System of coupled Stuart-Landau oscillators (or, complex Ginzburg-Landau oscillators): the Stuart-Landau model is known to exhibit amplitude death (aka quenching). Couldn't that be a likely alternative explanation for the sudden onset of synchrony, as presented in Fig. 5 (else, why not)?

References (e.g.): Koseska, A., Volkov, E., and Kurths, J. (2013). Oscillation quenching mechanisms: Amplitude vs. oscillation death. Physics Reports, 531(4), 173-199. Chabanol, M. L., Hakim, V., and Rappel, W. J. (1997). Collective chaos and noise in the globally coupled complex Ginzburg-Landau equation. Physica D: Nonlinear Phenomena, 103(1-4), 273-293.

Authors' response:

Overall, the Reviewer basically asks about potential alternative explanations underlying the phenomenon we report.

(i) D. Călugăru, et al. [13] raised a hypothesis that a sudden synchronization transition they observed emerges "*beyond phase reduction*" yet did not identify the detailed mechanism of the extreme nature of their observed synchronization transitions. We note that such a discontinuous synchronization transition may arise due to the following reasons: (a) First and traditionally, BZ reactions are modeled by oscillators beyond the standard phase reduction that inherently involve higher-order interactions. As explained above, such a system typically undergoes a discontinuous transition through a change in the criticality of a branching point bifurcation and then a delayed transition at a large coupling strength. (b) Secondly, it may also arise due to an *implicit* interplay between phase variables imposed with parameter disorder and other (here, amplitude) degrees of freedom. Then, it is reasonable to expect that the intrinsic parameter disorder of phase variables is absorbed or redistributed into other microscopic (here, amplitude) or possibly hidden macroscopic degrees of freedom, as we have pinned down (to the best of our knowledge, for the first time) in the current manuscript. However, this feature is not readily apparent in the system of BZ oscillators or other classical planar oscillators (van der Pol or Stuart-Landau oscillators) and, as such, has not been explicitly investigated to date.

In contrast, the complexified Kuramoto model, which also constitutes a coupled oscillator model beyond the standard phase reduction (technically speaking, defined on the surface of an infinite cylinder; $(x_\mu, y_\mu) \in [-\pi, \pi] \times \mathbb{R} \subset \mathbb{C}$), not only provides phenomenological observations, it also *explicitly* pins them down as genuinely extreme through an analytical approach and determines the core theoretical mechanism of extreme transitions to synchrony. Especially, we identify that and quantify how much the parameter disorder transfers from the phase variables to additional degrees of freedom. We provide a detailed explanation in Section "*Distinct nature of the transition*" in the revised manuscript.

(ii) Already a single (uncoupled) van der Pol oscillator undergoes a Canard explosion and so an explosive jump in its *amplitude*. This phenomenon involves a transition from a small-amplitude oscillatory state to a large-amplitude relaxation oscillatory state within an exponentially narrow range $\mathcal{O}(e^{-1/\epsilon})$ as a result of a singular perturbation *due to slow-fast time scales*. We consider a transition to synchrony which results from the coupling between

(two or more) units. Therefore, we do not consider a Canard explosion as an explanation for an extreme transition to synchrony observed here. We remark that Canard explosion could induce additional phenomena for coupled oscillators and if those are extreme could be an interesting topic for future research.

(iii) A system of coupled Stuart-Landau oscillators also exhibits an extremely ordered synchrony (Fig.5b) as a fixed point solution in an appropriate rotating reference frame, i.e., quenched oscillation. During synchronization transitions, parameter disorder (natural frequencies) of phase variables may be absorbed or redistributed into amplitude degrees of freedom or macroscopic observable defined by amplitude variables. This effect may indeed support extreme transitions but if true this is not a fact established as state of the art for phase-amplitude oscillators (please refer to our discussion in the manuscript) and, as the results in the manuscript imply, is not required to observe extreme transitions. Moreover, this mechanism is not readily revealed in a system of coupled Stuart-Landau oscillators.

Reviewer comment:

3. Finally, it may be interesting to note that systems with adaptive coupling and heterogeneous oscillator frequencies experience a widening of the synchronization region as the adaptive capability of the system is increased (effectively widening the associated Arnol'd tongue). It would appear as if the adaptive coupling may be able to absorb the effect of heterogeneity in oscillator frequencies, which would destroy synchrony in the absence of adaptive couplings. More generally, it would appear that oscillator models that effectively are higher dimensional, as is the case with the complexified Kuramoto model, have the ability to absorb heterogeneities in other variables.

References (e.g.): Ren, Q., and Zhao, J. (2007). Adaptive coupling and enhanced synchronization in coupled phase oscillators. *Phys. Rev. E* 76(1), 016207.

Authors' response:

We thank the Reviewer for this valuable comment and perspective. In the revised manuscript (Section "*Conclusion and scope.*"), we included a general discussion on an adaptive network or higher-dimensional oscillator models in relation to the redistribution of parameter disorder as a novel perspective for the interpretation of other explosive phenomena. We believe it provides a valuable additional perspective on the phenomenon we report and analyze.

Reviewer comment:

MINOR COMMENTS:

- *The notation clash for g (in Eq. 7) and the oscillator frequency, $g(\omega)$, clash.*

Authors' response:

In the revised manuscript, we use Q instead of g for Eq. (7) while keeping $g(\omega)$ for the natural frequency distribution.

Reviewer comment:

- *Why not use either α OR β as axis in Fig 2? They effectively show the same (I was at first confused and looking for incommensurate data types in the figure).*

Authors' response:

In the revised manuscript, we omit β -axis in Fig. 2 and also in Fig. 3c for readability.

Reviewer comment:

- *The time evolution of the complex order parameter, $Z(t)$, in the complex can be more revealing to see features. E.g., for the classical Kuramoto model, for time going to infinity, in the incoherent regime where $K < K_c$, the trajectory will be arbitrarily close to the origin, while it will never get close to the origin once $K > K_c$. Thus, there is a clear onset of synchrony visible when reporting $\min_t(|Z(t)|)$. Perhaps it is interesting to look at the oscillations reported in Fig .4d) in such a way.*

Authors' response:

We thank the Reviewer for this comment. Indeed, the time evolution of $Z(t)$ in the complex plane captures well the overall collective features, in particular for coupling below the transition. Yet, $\min_t(|Z(t)|)$ provides qualitatively the same picture. Figure R3 provides an example. We remark that in our manuscript, Figure 4d shows $r(t)$ for a range between 0 and about 0.2, indicating a low level of synchrony, whereas Figure 4e shows it on a range between 0 and 1, such that both cases are already well distinguished.

FIG. R3. (a) The modulus of the complex Kuramoto order parameter $\min_t(|Z(t)|)$ is depicted as a function of the coupling strength $|K|$. (b) The Kuramoto order parameter r as a function of the coupling strength $|K|$, imported from Fig. 1d in the main manuscript.

So, despite the general usefulness of $\min_t(|Z(t)|)$, we would like to keep the figure in the main manuscript as is, at least for the following reasons. Fig. 4 (d-f) shows not only the time evolution of the order parameter for synchrony but also reveals the frequency of oscillatory behavior ($\text{Im}\lambda \approx 3$) of the order parameter. Therefore, there is no qualitatively additional insight and we would need to introduce an additional observable. In summary, we thus hope you understand that we would like to keep the figure in the main manuscript as it is.

REPLIES TO THE COMMENTS OF REVIEWER #2

Reviewer comment:

Thank you for asking me to review the manuscript: “Extreme Synchronization Transitions”. This paper discusses transitions from disordered to synchronized state in networks of diffusively coupled oscillators. As their model of choice, the authors select a complexified variant of the prototypical Kuramoto model, with a unimodal frequency distribution. Each node in the network is represented by two dynamical variables (as opposed to the standard phase-only description). Nodes are then globally-coupled via a complex coupling strength $K = |K|e^{i\alpha}$.

The primary goal of the paper is to investigate synchrony transitions under variation of the coupling phase α , which adjusts weights of the coupling in the two dynamic variables. The authors show that at a critical value of α , the system undergoes a sharp transition from a state with disordered phases to a synchronized solution with a Kuramoto first order parameter near 1 for one of the two dynamic variables. The authors show through a combination of asymptotic analysis and numerical simulation that this transition is realizable in networks of finite size, in contrast to most studies of so-called “explosive synchrony transitions”. The authors show through numerical examples that the high levels of synchrony in the first dynamic variable are offset by an increased spread in the second dynamic variable. Finally, the authors show that similar types of explosive transition are observed in other model systems, including the Fitzhugh–Nagumo model, which they use as a representation of the Belousov–Zhabotinsky reaction.

Authors’ response:

We thank the Reviewer for all valuable comments and suggestions. We address the individual comments in the point-by-point replies below.

Reviewer comment:

Overall, the general findings and discussion around the explosive synchrony transitions are interesting, and raise important questions about how we should view synchrony in systems with partial observability. However, in spite of their interest, I am not sure about the novelty of the results in the present manuscript, particularly, because subsets of the authors have already published similar results in other journals, e.g. [1, 2]. I do not believe that the additions in the present manuscript are sufficiently novel with respect to the previous results to justify publication in a journal as high impact as Nature Communications. As such, in spite of my enthusiasm for the study, I cannot recommend it for publication.

References

- [1] M. Thümler, S. G. Srinivas, M. Schröder, and M. Timme, “Synchrony for weak coupling in the complexified Kuramoto model,” *Physical Review Letters*, vol. 130, no. 18, p. 187201, 2023.
- [2] S. Lee, L. Braun, F. Bönisch, M. Schröder, M. Thümler, and M. Timme, “Complexified synchrony,” *Chaos: An Interdisciplinary Journal of Nonlinear Science*, vol. 34, no. 5, 2024.
- [3] G. Ramesan, E. Shajan, and M. D. Shrimali, “Explosive synchronization induced by environmental coupling,” *Physics Letters A*, vol. 441, p. 128147, 2022. 2

Authors’ response:

We thank the Reviewer for the valuable comments. The Reviewer basically wonders about why and how the current work goes substantially beyond existing work.

First, as the Reviewer commented, our study offers an important perspective on how to reconsider the mechanism behind the onset and the extreme nature of discontinuous synchronization transitions. The current work provides new insights through analytical access that for the first time pins down the extreme nature of explosive phenomena, elucidates its scaling behavior upon varying a given system parameter, and reveals the core mechanism that the intrinsic disorder is redistributed to other variables. These aspects have remained unknown – and indeed unexplored – in previously reported explosive transitions, including experimental examples, models representing experimental features, typically studied

oscillator models beyond the standard phase reduction or phase models under higher-order interactions, and even our own prior numerical findings in Reviewer's Refs. [1-2]. In the below, we highlight the novel contributions of our findings in comparison to prior works.

(a) Ref. [1] (Thümler et al.) introduced complexification by analytic continuation for coupled phase oscillator models in the first place and points out that complexified variables may help understand the onset of synchronization phenomena because complex locked states exist also for weak (real) coupling. However, the "*extreme synchronization transitions*", disclosed in the current work, has not been reported there and (obviously) also not been explained. It newly enters the realm of synchronization phenomena, and is in fact not directly related to common explosive synchronization transitions. As demonstrated below, it is instead mediated by different bifurcation structures in the thermodynamic limit and as such qualitatively different, and furthermore emerges equally for finite, even small systems.

(b) Ref. [2] (Lee et al.) offers a general overview of the effects in coupled Kuramoto oscillators, where not only the variables (as in Ref. [1]) but also the parameters are complexified. One of the four data curves in Figure 5 of that work provided a numerical hint that something like extreme transitions might exist. However, the phenomenon has not been analytically pinned down nor systematically studied numerically before.

In the revised manuscript, we highlighted several major points about the novelty and the importance of the current study. Specifically, as listed in our *General Replies to all Reviewer's comments*, we distinguish four key dimensions: (i) the extremeness of transition and associated jump in order parameter to its theoretical maximum immediately past the transition point, (ii), its persistence in finite-size systems, (iii), the novel approach of obtaining analytic access by complexification, enabling us to pin down the phenomenon, and (iv), the identification of the core mechanism underlying extreme transitions.

Furthermore, the bifurcation inducing the onset of the extreme synchronization transition fundamentally differs from traditionally known discontinuous transitions or explosive phenomena. Whereas the latter emerges in the change of criticality of a branching point and an unstable branch bridges between the system's disordered and ordered states, in the extreme

transitions we report, in contrast, ordered and incoherent states are disconnected from each other and stable synchrony emerges via a Hopf bifurcation (Please see also Fig. R2 above for a reference illustration).

Reviewer comment:

Below, I have highlighted some other issues that the authors may wish to address.

- One of the main motivations for introducing the “extreme synchronization transition” is to reproduce the behavior of the Belousov-Zhabotinsky reactions reported in the inset of Fig. 1c. I acknowledge that the transition of the order parameter from low values to 1 is sharp, but not as sharp as the one observed in the FitzHugh–Nagumo oscillator network shown in the main Fig. 1c panel or in the complexified Kuramoto model in Fig. 1d. While “extreme synchronization transitions” are an interesting phenomenon, it remains unclear to me how much it captures the transition observed in the experiment better or worse than other transitions, as the comparison is mostly performed at a qualitative level.

Authors’ response:

We thank the Reviewer for these comments. Indeed, our study was motivated by experimental observations of extreme synchronization transitions whose underlying mechanisms remain unclear. Our goal is not to reproduce any specific experiment (or numerical observation) of a selected system. Unlike previous works that primarily reproduced discontinuous transitions (e.g., in Belousov-Zhabotinsky experiments), we show that extreme synchronization transitions represent a distinct and new class of phenomena. It may well also occur in classes beyond the Kuramoto model we employed to gain analytic access, points we discussed towards the end of the manuscript and illustrate by numerical examples.

Specifically, we provide both theoretical and numerical evidence that extreme behavior arises from the redistribution of intrinsic parameter disorder into additional variables’ degrees of freedom. Our analysis demonstrates how the order parameter can be precisely controlled by varying system parameters, and we establish the robustness of this mechanism across different natural frequency distributions, network topologies, and system sizes.

The revised manuscript (see Section "Distinct nature of the transition") elaborates on these points and offers a systematic theoretical framework for understanding these explosive synchronization transitions.

Reviewer comment:

- *I am wondering to what extent the term “extreme synchronization transition” is necessary as the phenomenon reported here refers to an explosive synchronization transition in a subset of variables. Perhaps “partial explosive synchronization transition” more accurately describes the observed dynamics, particularly since previous work [3] referred to the transition observed in Fig. 5a as an explosive transition?*

Authors’ response:

We thank the Reviewer for the valuable comments. Please let us emphasize that the main point is not that the order parameter of observed variables jumps discontinuously at all (as in many discontinuous and explosive types of transitions), but that it jumps extremely, i.e. very close to its maximum value of unity, immediately past the transition point.

In other words, in comparing the extreme synchronization transition studied in this article to previously studied explosive phenomena, the order parameter’s value, which jumps as close as possible to 1 (when varying the parameter β), makes the jump itself extreme, i.e., it approaches the maximum value, a feature not typically observed in explosive transitions so far. In this perspective, we believe that the term "extreme synchrony transition" is appropriate and prefer to retain it for the current study.

Reviewer comment:

- *For the Van der Pol oscillators, the choice of the natural frequencies (ω_μ) is very specific as given in the Supplementary Material and does not correspond to a Gaussian distribution as stated in the main text. Is the transition observed generically or for a specific choices of parameters?*

Authors' response:

Coupled Van der Pol oscillators indeed may exhibit extreme features in their transition to synchronization, as we illustrate on page 10 of the original manuscript. However, we also note that while the van der Pol oscillator meets conditions (i–ii) and exhibits extreme synchronization transitions, it might show a delayed transition at high K , missing properties (iii) and (iv) that are covered by the case study of our manuscript. This indicates that certain frequency distributions may reduce its extreme features.

For the complexified Kuramoto model specifically, extreme transitions emerge regardless of details in the natural frequency distribution, as confirmed by our simulations. The revised Supplementary Information includes results for uniform, Lorentzian, bimodal, and tri-modal Gaussian distributions, underscoring this robustness.

Reviewer comment:

- *It would be pertinent to discuss what types of dynamical features are common to systems that exhibit the “extreme synchronization transitions” described in the manuscript to help the reader understand when such behavior might be expected.*

Authors’ response:

We thank the Reviewer for pointing to this important open question. The extreme synchronization transition reported in this work arises from the absorption or redistribution of parameter disorder into other additional variables than phase state variables and lacks an unstable branch that bridges the desired synchrony and incoherence (cf. the traditional discontinuous transition, which stems from changes in the criticality of a branching point bifurcation).

So from the perspective of classes of general dynamical systems, extreme transitions require at least two variables per coupled unit so as to enable the absorption parameter disorder from some (e.g. phase) variables into other degrees of freedom. In the revised manuscript (Section "*Distinct nature of the transition*"), we discuss this point.

Reviewer comment:

- Is the fixed point specified in Eq. (4) the only fixed point that exists in the system?

Authors' response:

We thank the Reviewer for the thorough review. The fixed point called a complex locked state in Eq. (4) describes complexified synchrony. We numerically found that other fixed point solutions exist but they all are unstable and irrelevant to the synchronization phenomena. We pointed out this below eqn.(4) in the revised manuscript.

Reviewer comment:

- *Does the complexification of the coupling parameter have any effect of the discontinuous synchronization transitions observed in systems with bimodal frequency distributions?*

Authors' response:

In systems with a bimodal frequency distribution, a discontinuous synchronization transition is marked by (i) non-extreme synchrony and (ii) dependence on the thermodynamic limit. In contrast, complexifying the coupling parameter serves as the main key to redistributing intrinsic parameter disorder into imaginary variables, involving the extreme nature of synchrony in finite-size systems. Thus, we conclude that these two mechanisms are distinct.

REPLIES TO THE COMMENTS OF REVIEWER #3

Reviewer comment:

Authors' response:

We thank the Reviewer for their efforts and for the comments delivered jointly with one of the other Reviewers. We provide all point-by-point replies to the comments from all Reviewers.

REPLIES TO THE COMMENTS OF REVIEWER #4

Reviewer comment:

Comments on ‘Extreme Synchronization Transitions’ by Seungjae Lee, Lennart J. Kuklinski and Marc Timme.

The manuscript and Supplementary Material “Extreme Synchronization Transitions” present an intriguing study of a class of discontinuous collective dynamics in coupled oscillator systems. The authors explore extreme synchronization transitions, characterized by an abrupt leap from disordered states to near-complete synchrony, even in finite systems. Motivated by experimental observations in coupled Belousov-Zhabotinsky reactions, the study systematically investigates the dynamics and mechanisms underlying these transitions using coupled complexified Kuramoto oscillators as a model system.

The work is notable for uncovering distinct features of extreme transitions: dramatic changes in the order parameter, independence from thermodynamic limits, and a redistribution of parameter disorder among system variables. These findings are relevant to fundamental studies in nonlinear dynamics and practical applications in biological and engineered systems, including mitigating pathological synchrony in neural disorders or optimizing synchronization in power grids and robotic swarms.

I believe this manuscript is well-suited for publication in Nature Communications following revision. Below, I provide a detailed list of questions and suggestions to enhance the clarity and impact of the study.

Authors’ response:

We thank the Reviewer for their very positive judgements and their valuable comments and suggestions. We address the individual comments in the point-by-point replies below.

Reviewer comment:

Main Questions:

1. *Clarification on the Complexified Kuramoto Model: The Kuramoto model is widely recognized as a phase model derived from amplitude systems, with analogs in real-world applications. However, the concept of the complexified Kuramoto model remains somewhat unclear. Beyond the synchronization phenomena observed in Belousov-Zhabotinsky reactions, are there other real-world systems or scenarios where such a model has direct relevance? Providing concrete analogs or applications would help contextualize its significance. For instance, the manuscript provides examples of systems where extreme transitions might impact synchrony, such as neural diseases (e.g., Parkinson's or epilepsy) and power grids. However, it is unclear how the specific dynamics of the complexified Kuramoto model connect to these examples. Could the authors clarify whether the model presented in this study has direct applicability to these systems? If these examples are meant to illustrate general relevance rather than a direct connection, explicitly stating this would help set appropriate expectations for readers.*

Authors' response:

We thank the Reviewer for the insightful comments, which we used to improve the manuscript's clarity.

First, please let us emphasize that while the complexified Kuramoto model exhibits an extreme transition similar to the ones that motivated our study, it does not constitute a physical model of these other systems. In contrast, complexification by analytic continuation has helped to provide analytic access to the phenomenon of extreme synchrony. Such an approach has been highly useful before. For instance, complexifying thermodynamic parameters has helped build the foundations of phase transitions in statistical physics and complexifying operators have catalyzed the advent of PT-symmetric quantum mechanics.

The examples listed in the original manuscript such as neural disease and power-grid are complex systems that may cause serious dysfunction when an extreme or complete synchronization is achieved. Whereas other experimental and model studies (see Figure 1 in the

manuscript and our discussion section) are at least numerically or experimentally demonstrated to exhibit some form of extreme transition, and the complexified Kuramoto model we employ helps to pin down the phenomenon analytically and explain its core mechanism, other examples like the two systems mentioned above, remain unexplored (and are actually harder to investigate). Specifically for power grids, we are currently exploring how complexification may help to pin down conditions for extreme transitions, because these conditions have not been studied so far although the systems constitute essential infrastructural networks.

In line with these points, we intended to connect the examples to our study on the onset of or avoiding transitions to extreme system-wide order, upon varying a given system parameter. In the revised manuscript (Section "*Conclusion and scope.*"), we clarified why the phenomenon of extreme synchrony transitions studied in this work may have general relevance to the applications listed.

Reviewer comment:

2. *Thermodynamic Limit and Finite System Size:* The authors emphasize that extreme synchronization transitions occur in finite systems, without requiring the thermodynamic limit. However, not all discontinuous synchronization transitions rely on the thermodynamic limit. For example, models with bimodal frequencies, Janus oscillators, or setups for explosive synchronization can exhibit similar phenomena without such requirements. Given these precedents, what is the specific theoretical or practical importance of emphasizing this distinction in the context of extreme synchronization? This clarification would strengthen the manuscript's conceptual contribution.

Authors' response:

We thank the Reviewer for this comment upon which we slightly adapted the manuscript presentation. In our perception, most of the community working on synchronization phenomena still thinks of continuous transitions as the typical case and of discontinuous transitions as exceptions. The work by Kühn and Bick (C. Kühn and C. Bick, *Science Advances* **7**, abe3824 (2021).) have helped to gradually change that view. For discontinuous and explosive transitions, again, most of the community focus is on the thermodynamic limit. Cases, where there is evidence for discontinuous transitions in finite systems almost exclusively rely on numerical observations, in particular for explosive transitions more generally (i.e. beyond synchronization).

So overall, finite-size discontinuous transitions are being perceived as rather an exception. This is why we believe it is worth emphasizing our observation of *extreme* transitions in finite systems: They constitute clear examples of systems with transitions that are all, (i) discontinuous, (ii) extreme, and (iii) occurring at finite and even small system sizes.

Moreover, the class of systems we analyze in the current manuscript features analytical accessibility for finite-size systems, enabling us to pin down the features (i-iii) with certainty. More generally, as also emphasized by Reviewer #1, they open up broad applicability to achieve or prevent extreme transition features in complex systems that are finite in the real world in any case. In the revised manuscript, we put some more emphasis on this perspective. Thank you again for raising the question.

Reviewer comment:

Minor Questions:

1. *Presentation of the Kuramoto Model: In the manuscript, the authors discuss the Kuramoto model as a foundational concept before introducing the complexified Kuramoto model. While this sets up the context, the figures illustrating results (e.g., those based on coupling strength) appear before the governing equations and definitions are introduced. This presentation could be confusing for readers unfamiliar with the Kuramoto model or its modifications. Could the authors consider including the equations and key definitions earlier in the text, ideally before presenting the simulation results?*

Authors' response:

We thank the Reviewer for this comment and their thorough consideration. We agree that the proposed structure may help to bring across the mathematical details for readers who are not familiar with the Kuramoto model. However, we would like to maintain the current structure, because we think that vividly highlighting the distinction between transition *phenomena* is more essential to the current work than the details of the models or experimental *systems* these phenomena occur in. We thank you for your understanding.

Reviewer comment:

2. *Definition of the Phase-Like State Variable: In calculating the order parameter, the manuscript references the state vector $x_\nu \in \mathbb{R}$. However, it is not entirely clear what is meant by the “phase-like” state variable mentioned in the manuscript. If x_ν is not the conventional phase variable used in the Kuramoto framework, could the authors clarify its meaning and how it relates to the phases typically used in synchronization studies? Providing a concise definition of x_ν and how it differs from or extends conventional phase variables would enhance clarity. This question might also be relevant to the broader question about the complexified Kuramoto model (see Main Question 1).*

Authors' response:

We thank the Reviewer for the valuable comment. Indeed, the real part x_μ of a complexified Kuramoto oscillator is identical to the phase variable of the original, real-variable model,

as common in the studies on synchronization. We have revised the manuscript (below eqn. (1)), and now use "phase variables $x_\mu \in \mathbb{T}$ " instead of "phase-like variables $x_\mu \in \mathbb{R}$ ".

Reviewer comment:

3. Legend Transparency in Figure 1: In Figure 1, the legend uses varying transparency levels to distinguish system sizes, with transparency increasing from large systems to small ones. This choice is not immediately intuitive and may be confusing for readers. Could the authors clarify the rationale behind this design choice? Alternatively, adopting a more conventional style, such as distinct colors or line styles, might improve readability.

Authors' response:

We thank the Reviewer for this comment. We fully agree with your comment. In the revised manuscript, we changed the figure to make it more readable. In the revised Fig. 1, we use filled dots for $N = 8$, dashed curves for $N = 64$, and solid curves for $N = 2048$.

Reviewer comment:

4. Color Discrepancy in Figure 3: The caption of Figure 3 refers to phase and other variables using purple and orange colors, respectively. However, these colors do not appear in the plot, which may lead to confusion. Could the authors verify and correct the caption or the figure itself to ensure consistency?

Authors' response:

We thank the Reviewer for this comment. In the revised manuscript, the Figure 3 caption is corrected.

Reviewer comment:

The manuscript provides significant insights into extreme synchronization transitions and their implications across various systems. The theoretical framework is rigorous, and the supplementary materials offer an excellent resource for readers interested in additional technical details. However, addressing the questions above, particularly those related to the real-world relevance of the complexified Kuramoto model, would further enhance the clarity and impact of the work.

I recommend publication following these revisions, as the findings are likely to attract broad interest within the readership of Nature Communications.

Authors' response:

We thank the Reviewer again for reviewing our manuscript and for their positive judgment of the topic, content and presentation of our results.

Thank you for asking me to review the manuscript: “Extreme Synchronization Transitions”. This paper discusses transitions from disordered to synchronized state in networks of diffusively coupled oscillators. As their model of choice, the authors select a complexified variant of the prototypical Kuramoto model, with a unimodal frequency distribution. Each node in the network is represented by two dynamical variables (as opposed to the standard phase-only description). Nodes are then globally-coupled via a complex coupling strength $K = |K|e^{i\alpha}$.

The primary goal of the paper is to investigate synchrony transitions under variation of the coupling phase α , which adjusts weights of the coupling in the two dynamic variables. The authors show that at a critical value of α , the system undergoes a sharp transition from a state with disordered phases to a synchronized solution with a Kuramoto first order parameter near 1 for one of the two dynamic variables. The authors show through a combination of asymptotic analysis and numerical simulation that this transition is realizable in networks of finite size, in contrast to most studies of so-called “explosive synchrony transitions”. The authors show through numerical examples that the high levels of synchrony in the first dynamic variable are offset by an increased spread in the second dynamic variable. Finally, the authors show that similar types of explosive transition are observed in other model systems, including the Fitzhugh–Nagumo model, which they use as a representation of the Belousov–Zhabotinsky reaction.

Overall, the general findings and discussion around the explosive synchrony transitions are interesting, and raise important questions about how we should view synchrony in systems with partial observability. However, in spite of their interest, I am not sure about the novelty of the results in the present manuscript, particularly, because subsets of the authors have already published similar results in other journals, e.g. [1, 2]. I do not believe that the additions in the present manuscript are sufficiently novel with respect to the previous results to justify publication in a journal as high impact as Nature Communications. As such, in spite of my enthusiasm for the study, I cannot recommend it for publication.

Below, I have highlighted some other issues that the authors may wish to address.

- One of the main motivations for introducing the “extreme synchronization transition” is to reproduce the behavior of the Belousov-Zhabotinsky reactions reported in the inset of Fig. 1c. I acknowledge that the transition of the order parameter from low values to 1 is sharp, but not as sharp as the one observed in the FitzHugh–Nagumo oscillator network shown in the main Fig. 1c panel or in the complexified Kuramoto model in Fig. 1d. While “extreme synchronization transitions” are an interesting phenomenon, it remains unclear to me how much it captures the transition observed in the experiment better or worse than other transitions, as the comparison is mostly performed at a qualitative level.
- I am wondering to what extent the term “extreme synchronization transition” is necessary as the phenomenon reported here refers to an explosive synchronization transition in a subset of variables. Perhaps “partial explosive synchronization transition” more accurately describes the observed dynamics, particularly since previous work [3] referred to the transition observed in Fig. 5a as an explosive transition?

- For the Van der Pol oscillators, the choice of the natural frequencies (ω_μ) is very specific as given in the Supplementary Material and does not correspond to a Gaussian distribution as stated in the main text. Is the transition observed generically or for a specific choices of parameters?
- It would be pertinent to discuss what types of dynamical features are common to systems that exhibit the “extreme synchronization transitions” described in the manuscript to help the reader understand when such behavior might be expected.
- Is the fixed point specified in Eq. (4) the only fixed point that exists in the system?
- Does the complexification of the coupling parameter have any effect of the discontinuous synchronization transitions observed in systems with bimodal frequency distributions?

References

- [1] M. Thümler, S. G. Srinivas, M. Schröder, and M. Timme, “Synchrony for weak coupling in the complexified kuramoto model,” *Physical Review Letters*, vol. 130, no. 18, p. 187201, 2023.
- [2] S. Lee, L. Braun, F. Bönisch, M. Schröder, M. Thümler, and M. Timme, “Complexified synchrony,” *Chaos: An Interdisciplinary Journal of Nonlinear Science*, vol. 34, no. 5, 2024.
- [3] G. Ramesan, E. Shajan, and M. D. Shrimali, “Explosive synchronization induced by environmental coupling,” *Physics Letters A*, vol. 441, p. 128147, 2022.